mRNA sequencing reveals the distinct gene expression and biological functions in cardiac fibroblasts regulated by recombinant fibroblast growth factor 2

http://orcid.org/0000-0001-8602-9629 Sun Changye 1 sunchangye-boy@163.com
Bai Mengru 1
Jia Yangyang 1
Tian Xiangqin 1
Guo Yonglong 2
Xu Xinhui 1
Guo Zhikun 1 gzk@xxmu.edu.cn
1 Henan Key Laboratory of Medical Tissue Regeneration, Xinxiang Medical University , Xinxiang, Henan , China
2 Department of Cardiology, The First Affiliated Hospital, Xinxiang Medical University , Xinxiang, Henan , China
Gould Gwyn
Electronic publication date: 2023 Jul 19
Publication date: 2023
Volume: 11
Electronic Location ID: e15736
Received 2023 Feb 9; Accepted 2023 Jun 20
Copyright: © 2023 Sun et al.
Copyright year: 2023
Copyright holder: Sun et al.
License: This is an open access article distributed under the terms of the Creative Commons Attribution License, which permits unrestricted use, distribution, reproduction and adaptation in any medium and for any purpose provided that it is properly attributed. For attribution, the original author(s), title, publication source (PeerJ) and either DOI or URL of the article must be cited.
License URL: https://creativecommons.org/licenses/by/4.0/

Keywords: Fibroblast growth factor 2 (FGF2), Cardiac fibroblasts (CFs) activation, mRNA sequencing, Extracellular matrix (ECM) organization, Actin filament organization

Funding: Xinxiang Medical University 300-505232 Henan Province Science and Technology Innovation Talent Program 212102310873 This study was funded by the Xinxiang Medical University (300-505232) and the Henan Province Science and Technology Innovation Talent Program (212102310873). The funders had no role in study design, data collection and analysis, decision to publish, or preparation of the manuscript.

==============================
After myocardial injury, cardiac fibroblasts (CFs) differentiate into myofibroblasts, which express and secrete extracellular matrix (ECM) components for myocardial repair, but also promote myocardial fibrosis. Recombinant fibroblast growth factor 2 (FGF2) protein drug with low molecular weight can promote cell survival and angiogenesis, and it was found that FGF2 could inhibit the activation of CFs, suggesting FGF2 has great potential in myocardial repair. However, the regulatory role of FGF2 on CFs has not been fully elucidated. Here, we found that recombinant FGF2 significantly suppressed the expression of alpha smooth muscle actin (α-SMA) in CFs. Through RNA sequencing, we analyzed mRNA expression in CFs and the differently expressed genes regulated by FGF2, including 430 up-regulated genes and 391 down-regulated genes. Gene ontology analysis revealed that the differentially expressed genes were strongly enriched in multiple biological functions, including ECM organization, cell adhesion, actin filament organization and axon guidance. The results of gene set enrichment analysis (GSEA) show that ECM organization and actin filament organization are down-regulated, while axon guidance is up-regulated. Further cellular experiments indicate that the regulatory functions of FGF2 are consistent with the findings of the gene enrichment analysis. This study provides valuable insights into the potential therapeutic role of FGF2 in treating cardiac fibrosis and establishes a foundation for further research to uncover the underlying mechanisms of CFs gene expression regulated by FGF2.

Introduction

Myocardial fibrosis is a common consequence of various myocardial injuries, leading to multiple cardiac diseases such as cardiac hypertrophy and arrhythmia (Talman & Ruskoaho, 2016; Travers et al., 2016). Myocardial injury triggers the activation of cardiac fibroblasts (CFs), which are transformed into active myofibroblasts (Forte et al., 2020; Fu et al., 2018). Myofibroblasts characterized by expression of alpha smooth muscle actin (α-SMA) play an essential role in the repair of damaged myocardium by producing extracellular matrix (ECM) proteins (Fu et al., 2018; Furtado et al., 2016; Ma et al., 2017; Travers et al., 2016). However, overexpression and secretion of ECM proteins in the damaged tissue caused cardiac fibrosis seriously affecting the physiological functions of the heart (Ma et al., 2017; Talman & Ruskoaho, 2016; Travers et al., 2016). Effective suppression of CFs activation is the key to inhibit ECM deposition, which is of great significance for the treatment of myocardial fibrosis (Sun et al., 2022).

Fibroblast growth factor (FGF) was first isolated from bovine pituitary gland and was found to stimulate DNA synthesis in 3T3 fibroblasts (Gospodarowicz, 1975). The subsequent studies revealed that FGFs could bind to its receptors and co-receptors, heparan sulfate, and regulate signal transduction between cells, which play extremely important roles in embryonic and organism development, homeostasis, tissue repair and diseases (Beenken & Mohammadi, 2009; El Agha, Seeger & Bellusci, 2017; Itoh & Ornitz, 2011; Sun et al., 2016). A couple of studies have found that several recombinant FGFs, especially FGF2, can inhibit activation of fibroblasts derived from multiple organs, including lung, skin, heart and cornea, and the expression of collagens in these fibroblasts was also decreased by FGF2 (Gallego-Munoz et al., 2017; Koo et al., 2018; Liguori et al., 2018; Ramos et al., 2006; Santiago et al., 2014b; Svystonyuk et al., 2015). Recombinant FGF2, as a gene-engineered protein drug, has been used to clinical treatment in China and Japan (Nunes et al., 2016). Clinical studies have shown that FGF2 enhances tissue healing and angiogenesis without promoting skin tissue fibrosis and severe scar formation (Nunes et al., 2016). FGF2 had been applied to myocardial injury in several animal studies to promote tissue repair and angiogenesis, but its role in CFs activation and ECM organization has not been fully investigated (Liao et al., 2009, 2007; Sun et al., 2022).

In myocardial tissue, two subtypes of FGF2, high molecular weight FGF2 (HMW-FGF2) and low molecular weight FGF2 (LMW-FGF2), were determined (Jiang et al., 2007; Santiago et al., 2014b). It was found that HMW-FGF2 could promote cardiac hypertrophic and CFs activation, while LMW-FGF2 did not promote CFs activation and has certain protective functions to the heart (Jiang et al., 2007; Liao et al., 2007; Santiago et al., 2014a). The recombinant LMW-FGF2 protein drug has been reported to suppress activation of CFs, which is of great significance for the repair and remodeling of injured myocardium (Liao et al., 2009; Liguori et al., 2018; Svystonyuk et al., 2015). Moreover, it was found that the increase of LMW-FGF2 by overexpression or drug treatment could reduce the degree of fibrosis in multiple tissues, e.g., the liver and lung (Koo et al., 2018; Pan et al., 2015). Despite these findings, the full scope of genes and functions regulated by FGF2 in CFs remains unclear. In this study, we utilized mRNA sequencing to identify the role of FGF2 in the regulation of CFs gene transcription, with the aim of shedding light on the regulatory effects of FGF2 on CFs.

Materials and Methods

Materials

Recombinant histidine-tagged LMW-FGF2 protein (18 kDa) was used as FGF2 protein drug in this study. Recombinant FGF2 was expressed by the prokaryotic expression system (E. coli strain: BL21; Transgen Biotech, Beijing, China) and purified by heparin affinity gel (Bio-Rad, Hercules, California, USA) and nickel affinity gel (GE Healthcare Life Sciences, Chicago, Illinois, USA), as described previously (Sun et al., 2015, 2019). For primary rat cardiac fibroblast isolation and culture, the following materials were used: collagenase type II (Gibco, Thermo Fisher Scientific, Shanghai, China), Dulbecco’s modified Eagle’s medium (DMEM) containing 1,000 mg/L glucose and 4 mM L-Glutamine (HyClone Laboratories, Logan, UT, USA), fetal bovine serum (FBS, Gibco, Billings, Montana, USA), phosphate buffered saline (PBS, Solarbio, Beijing, China), trypsin (Solarbio, Beijing, China), and penicillin-streptomycin (Solarbio, Beijing, China), cell culture dishes and plates (Corning, Oneonta, NY, USA). For RNA isolation and quantification, RNAiso Plus and TB Green Premix Ex Taq II were purchased from Takara Biomedical Technology (Beijing, China), and RevertAid First Strand cDNA Synthesis kit was ordered from Thermo Fisher Scientific. Antibodies, including Anti-DDR2 antibody (ab221812), Anti-alpha smooth muscle Actin (ab7817), anti-collagen type III antibody (ab7778), and Alexa Fluor® 488 conjugated antibodies (ab150113 for anti-mouse and ab150077 for anti-rabbit) were purchased from Abcam (Shanghai, China). Propidium iodide (PI, Beyotime, Shanghai, China) was used to stain nuclei and 5(6)-Carboxyfluorescein diacetate N-succinimidyl ester (CFSE, Sigma, Steinheim, Germany) was applied to stain the cells for fluorescence imaging.

CFs isolation and culture

Neonatal Sprague-Dawley rats (10–15 rats, 3 days old) were obtained from Experimental Animal Centre of Xinxiang Medical University and the animal studies were approved by the Xinxiang Medical University Animal Supervision Committee (approval no. XYLL-2020078). The neonatal rats were decapitated using sterile scissors, and hearts were extracted from the body and immediately washed with DMEM medium. The isolated hearts were cut into small pieces (approximately 1 mm3). The small tissues were digested by collagenase type II (0.05% w/v) and trypsin (0.06% w/v) in DMEM for 10 min at 37 °C, and the supernatant was transferred into DMEM with 10% (v/v) FBS. The digestion was repeated for three times. The cells and undigested tissues were separated by filtration and the isolated primary cells were pelleted by centrifugation at 1,000 × g for 5 min. The primary cells were seeded into a tissue culture plate with DMEM containing 10% FBS and 1% (v/v) penicillin/streptomycin for fibroblast adhesion, and after 45 min the unattached cells were removed to purify CFs. CFs were subcultured for three passages (P0-P3) to detected their differentiation in vitro. The primary CFs were split into six well plates for RNA isolation and 48 well plates for immunofluorescence. To determine the effect of FBS on CFs activation, the cells were cultured with DMEM containing 0%, 2.5%, 5% and 10% (v/v) FBS for 24 h. The bright field images were captured by an inverted microscope (T-DH; Nikon, Tokyo, Japan).

Treatment CFs with FGF2

Different concentrations of FGF2 (1, 3, 10, 30, 100 ng/mL) were added into DMEM with 2.5% FBS to determine their regulatory functions on CFs growth and activation. Subsequently, 30 ng/mL FGF2 was used to treat CFs for determination of its functions. As CFs attached to the plate, CFs in control group (CFs-CON) were treated with DMEM containing 2.5% FBS and CFs in FGF2-treated group (CFs-FGF2) were treated with DMEM containing 2.5% FBS and 30 ng/mL FGF2. The treated cells were collected for mRNA quantification and immunofluorescence imaging. For detection of the expression of collagen type III, CFs-CON and CFs-FGF2 were incubated for 5 days and the culture medium was refreshed every 2 days.

Immunofluorescence imaging

The cultured CFs in 48 well plates were fixed with 4% (w/v) paraformaldehyde (PFA) for 15 min and permeabilized with 0.3% (v/v) Triton X-100 for 15 min at room temperature. The cells were blocked with goat serum for 1 h. The purity of isolated cells was quantified by detecting the expression of discoidin domain receptor 2 (DDR2, antibody concentration: 2 μg/mL). To detect CFs activation, expression of α-SMA (antibody concentration: 1 μg/mL) was determined with immunofluorescence staining. Expression of collagen type III (antibody concentration: 2 μg/mL) was detected to validate the mRNA sequencing result. The fluorescent secondary antibodies (1:500 dilution) were applied to display fluorescence for imaging. Cell nuclei were stained with PI. Fluorescent images were captured by a laser scanning confocal microscope (FV1000; Olympus, Tokyo, Japan). Each imaging experiment was performed with three biological replicates and two technical replicates.

Real-time polymerase chain reaction

CFs-CON and CFs-FGF2 were collected with RNAiso Plus, and total RNA was extracted and quantified by Nanodrop ND-100 (Thermo Fisher Scientific, Shanghai, China). Reverse transcription was performed according to RevertAid First Strand cDNA Synthesis kit protocol. The primers used in this study were designed with Oligo 6 (Molecular Biology Insights, Cascade, CO, USA), and the primer synthesis was performed by GENEWIZ (Suzhou, China). The sequences of primers are listed in Table S1. RT-PCR was carried out with TB Green Premix Ex Taq II on a QuantStudioTM 7 Flex Real-Time PCR system (Applied Biosystems, Life Technologies, Waltham, MA, USA). Glyceraldehyde-3-phosphate dehydrogenase (GAPDH) was applied as the endogenous standard for subsequent analysis. Three biological replicates were performed for RT-PCR experiments.

RNA sequencing and data analysis

CFs-CON and CFs-FGF2 were collected with Trizol and mRNA was extracted by RNeasy Mini Kit (Qiagen, Shanghai, China). The total RNA was quantified and qualified by Nanodrop and Agilent 2100 Bioanalyzer (Agilent Technologies, Palo Alto, CA, USA). Library preparation and RNA sequencing with Illumina NovaSeq 6000 were processed by GENEWIZ. HISAT2 (v2.0.1) was used to align the collected gene sequences to Rattus genome (Rattus_norvegicus.Rnor_6.0.dna.toplevel) and HTSeq (v0.6.1) estimated gene expression levels from the pair-end clean data. The mapped reads were normalized to fragments per kilobase of exon model per million mapped fragments (FPKM) for comparison of gene expression levels. Three biological replicates were used in each group and the Pearson correlation coefficient is over than 0.92 (Fig. S1). The statistical power of this experimental design, calculated in RNASeqPower is 0.929. DESeq2 Bioconductor package was performed to analyze the difference of gene expression. Volcano plot and GeneExpression-FoldChange plot (scatter plot of gene expression level against fold change) were displayed in Matlab (2019a) to visualize the significantly regulated genes (log2(foldchange) ≥ 0.75 or log2(foldchange) ≤ −0.75) with limited expression levels (log10(baseMean) ≥ 2) and q values (q < 0.05, p values adjusted by false discovery rate (FDR) method). The MATLAB code for volcano plot and GeneExpression-FoldChange plot can be downloaded from GitHub at https://github.com/hscsun/ScatterFoldChanges.git. Gene ontology (GO) enrichment was analyzed and displayed with clusterProfiler package in R (3.6) and Metascape (Wu et al., 2021b; Zhou et al., 2019).

Cell adhesion assay

CFs were incubated in a 6-well plate at a density of 5 × 105 cells/well with and without FGF2 for 24 h. Both CFs-CON and CFs-FGF2 were trypsinized for 30 s and immediately fixed with 4% PFA. After washing with PBS, the cells were stained with CFSE for 1 h at room temperature. Nuclei were stained with PI. Images were taken by a laser scanning confocal microscope.

Results

CFs isolation and culture in vitro

Cardiac tissues were dispersed to isolate cardiac cells and the isolated cells were seeded into a plate. After 45 min, the unattached cells were removed by medium replacement to purify CFs and the purified CFs were subcultured to detect their activation in vitro (Fig. 1A). The immunofluorescence result shows the isolated cells express DDR2 indicating the isolated cells are mainly cardiac fibroblasts (Fig. 1B). Bright field imaging shows the morphology of primary CFs is more solid and after one passaging the morphology of CFs is relatively flat (Fig. 1C), which is consistent with the previous findings (Santiago et al., 2010). The immunofluorescence results also show the expression of α-SMA in CFs was increased as cultured in vitro (Fig. 1D). Since the isolated CFs were rapidly activated during incubation in vitro, the first passage of CFs was applied for drug treatment and determination of their biological activities.

Figure 1 CFs isolation and activation cultured in vitro.

(A) Cartoon model of CFs isolation and culture in vitro. (B) Determination of CFs by DDR2 staining. (C) Morphological change of CFs P0 and CFs P1 in bright filed. (D) Expression of α-SMA in CFs from P0 to P3. Scale bar: 200 µm.

CFs were treated with DMEM containing 0%, 2.5%, 5% and 10% of FBS for 24 h and fixed with 4% PFA. Expression of α-SMA of fixed CFs was determined to reveal the regulatory effect of FBS on CFs activation. The results show that CFs treated with 2.5% FBS express more α-SMA than CFs treated with 5% and 10% FBS (Fig. 2), suggesting CFs activation was more obvious in low serum medium. The nucleic acid staining of CFs treated without FBS was weaker than that of other groups, indicating the essential role of FBS in DNA synthesis (Fig. 2). To reduce the effect of FBS on CFs gene expression, DMEM containing 2.5% FBS was used for treatment of CFs with FGF2.

Figure 2 Activation of CFs regulated by FBS.

Expression of α-SMA in CFs cultured with different concentrations of FBS. Scale bar: 200 µm.

Regulatory effect of FGF2 on CFs activation

CFs were treated with induction medium containing various concentrations of FGF2 (0, 1, 3, 10, 30, 100 ng/mL) for 24 h to determine the effects of FGF2 on CFs activation. Expression of α-SMA protein was detected to analyse the level of CFs activation. It shows that incubation with FGF2 (3–30 ng/mL) could significantly reduce the expression of α-SMA in CFs and 30 ng/mL FGF2 displayed a better suppressive effect than 3–10 ng/mL FGF2 (Fig. 3A). RT-PCR results show that FGF2 could significantly supress the expression of myofibroblast marker genes (Acta2 and Tagln) and ECM genes (Col3 and Ccdc80) and higher concentration was more efficient (Fig. 3B). These indicate that FGF2 is a potential protein drug to inhibit CFs activation and the inhibitory effect is sensitive and concentration-dependent. Thus, 30 ng/mL FGF2 was applied to treat CFs to detect its effect on CFs, and the bright field imaging results show that the cellular morphologies of CFs-FGF2 are slenderer than CFs-CON, indicating FGF2 has a significant regulatory effect on CFs (Fig. 3C).

Figure 3 Inhibition of CFs activation by FGF2.

(A) Expression of α-SMA in CFs treated with different concentrations of FGF2. (B) RT-PCR quantification of mRNA expression of Acta2, Tagln, Col3 and Ccdc80 in CFs treated with different concentrations of FGF2. Three biological replicates were performed. (C) Bright field images of CFs treated without and with 30 ng/mL FGF2. Scale bar: 200 µm.

Sequencing analysis of CFs gene expression

Highly expressed genes in CFs

To explore the regulation of FGF2 on gene expression, CFs-FGF2 and CFs-CON respectively incubated with and without FGF2 were collected for mRNA sequencing. The mapped genes in CFs-CON were sorted by gene expression level (FPKM), and it shows that the top 60 highly expressed genes contain many genes of ECM proteins, e.g., Col1a1, Col3a1, Sparc, Fn1, Actb, Acta2, Tagln, Tpm4 and Myl6 (Fig. 4A). Some of these are up-regulated or down-regulated by FGF2, e.g., Actb, Acta2, Col3a1, Sparc and Ctgf (Fig. 4A). Enriched functions of the top 500 highly expressed genes were analysed by Metascape. The results demonstrate that the high expressed genes are enriched in ribosome, metabolic process, homeostasis, ECM organization, wounding, supramolecular fiber organization, cell motility, cell-substrate adhesion, smooth muscle contraction, etc., which are consistent with the biological activities of fibroblasts (Fig. 4B). In these enriched functions, ECM organization, collagen metabolic process, wounding, supramolecular fiber organization, cell-substrate adhesion and focal adhesion are strongly related functions, which are involved in ECM organization and fibrosis (Fig. 4B).

Figure 4 RNA sequencing revealed the high expression genes in CFs.

(A) Top 60 highly expressed genes in CFs-CON. (B) Enriched functions of top 500 high expression genes in CFs-CON were analysed and clustered with Metascape. The colours of functions are consistent with the colours of clusters. Three biological replicates were performed for RNA sequencing.

Differentially expressed genes (DEGs) regulated by FGF2

The mapped genes were scattered into five groups, low expression genes (log10(baseMean) < 2), not significant genes (q ≥ 0.05), low fold change genes (−0.75 < log2(fold change) < 0.75), up-regulated genes and down-regulated genes. Of these, 430 up-regulated genes and 391 down-regulated genes are considered to be the differently expressed genes regulated by FGF2 (Figs. 5A and 5B). The heatmap of differently expressed genes indicates that gene expressions of the repeated samples are consistent, while the expression levels of CFs-CON and CFs-FGF2 are obviously different (Fig. 5C). To analyse the effect of FGF2 on CFs biological functions, the differently expressed genes were applied to GO enrichment analysis. The enriched functions of biological process (BP), molecular function (MF) and cellular component (CC) were clustered and the major functions with high gene counts and low similarity were displayed to recognise the significantly regulated functions (Fig. 5D). Angiogenesis, protein kinase activity, cell-substrate adhesion, actin filament organization and extracellular matrix organization are involved in BP group (Fig. 5D). The MF group contains functions including receptor regulator activity, enzyme activator activity, actin binding, collagen binding, etc. (Fig. 5D). Differently expressed genes related to cell components are enriched in ECM, adherens junction, receptor complex, membrane microdomain, actin cytoskeleton, site of polarized growth and apical part of cell (Fig. 5D). Actin filament organization is enriched in all these three groups, and both ECM organization and receptor complex are also strongly enriched functions. These indicate FGF2 significantly regulates transcription of receptor-ligand genes, ECM component genes and actin cytoskeleton genes. To determine the accuracy of the mRNA sequencing, we quantified some significantly regulated genes, including Acta2, Col3a1, Ccdc80, Eln, Gas6, Itgb1, Tagln and Thbs2, and the results show that the RT-PCR results are consistent with mRNA sequencing results (Fig. 5E), suggesting the mRNA sequencing result is reliable and could be used to analyse the effects of FGF2 on CFs gene expression and biological function.

Figure 5 Differentially expressed genes (DEGs) in CFs regulated by FGF2.

(A and B) DEGs were clustered into five groups as listed. Fold change of expression levels were plotted against adjusted p value (q value) (A) and against baseMean (B). (C) Heatmap of expression levels of DEGs in each sample. (D) GO enrichment analysis of DEGs. (E) RT-PCR validation of DEGs genes. Three biological replicates were performed for RNA sequencing and RT-PCR experiments.

The principle enriched functions regulated by FGF2 in cellular component cluster

Cellular component enrichment presents the enriched genes and functions in cellular component organization. The GO enrichment network of cellular component shows that the highly enriched functions are involved in membrane organization (pink labelled functions), pericellular and extracellular matrix organization (blue labelled functions) and cytoskeleton organization (brown labelled functions) (Fig. 6A). Of these, ECM, adherens junction, actin cytoskeleton, which are fundamental for CFs activation and tissue fibrosis. In actin cytoskeleton group, many genes were strongly decreased, e.g., Myh7a, Myh9a, Myh10, Myh11, Cald1, Lmod1, and Daam1 (Fig. 6B). A large number of genes related to ECM, including genes of ECM protein (Col3a1, Col11a1, Cthrc1, Postn, Eln, Fmod, Ccdc80, Gpc4, Gpc6, etc.) and genes of signalling proteins (Wnt4, Tgfb2, Tgfb3, etc.), were decreased, although some genes were increased, e.g., Col7a1, Mmp16 and Col8a1 (Fig. 6B). In adherens junction cluster, a bunch of genes, e.g., Plekha7, Jag1, Itgbl1 and a bunch of genes, e.g., Plekha7, Jag1 and Itgb1, were decreased (Fig. 6B).

Figure 6 Enriched functions and relevant genes in cellular component cluster.

(A) Network of principle enriched functions in cellular component cluster. Membrane organization relevant functions were colored in pink; extracellular matrix relevant functions were colored in blue and cytoskeleton organization relevant functions were colored in brown. (B) Display of genes related to actin cytoskeleton, adherens junction and extracellular matrix.

The differently expressed genes were also enriched in clusters of membrane microdomain, basolateral plasma membrane and transport vesicle, and these functions were up-regulated by the relevant genes (Fig. S2). Gene set enrichment analysis (GSEA) results indicate actin cytoskeleton was suppressed by FGF2, while axon guidance was promoted, and many genes in cell leading edge function were increased as listed suggesting FGF2 might promote the polarized growth of CFs (Fig. S3).

GSEA was also applied to quantify effects of FGF2 on ECM organization and cell adhesion of CFs. The analysis results indicate both functions were down-regulated by FGF2 (p < 0.05), suggesting FGF2 could suppress these cell functions (Fig. 7A). Expression of collagen type III in CFs-CON and CFs-FGF2 were stained, and it shows collagen type III could be expressed and accumulated by CFs-CON, while it was not strongly accumulated in CFs-FGF2 (Fig. 7B). These demonstrated that activated CFs expressed large amounts of collagen and induced collagen accumulation, but FGF2 could to some extent suppress this expression and accumulation. To determine the adhesion of CFs to substrate, the cells were stained with CFSE to visualize the adhesion area of the cells. The results show that the adhesion areas of CFs-FGF2 are much smaller than those of CFs-CON, indicating FGF2 could decrease the adhesion of CFs to substrate (Fig. 7C).

Figure 7 Regulation of ECM organization and cell adhesion.

(A) GSEA plots indicate FGF2 decreased ECM organization and cell-substrate adhesion in CFs. (B) Expression of collagen type III in CFs-CON and CFs-FGF2 was determined by immunofluorescence staining. (C) CFs were stained with CFSE to display cell adherent areas. Scale bar: 200 µm.

Discussion

In this study, we found that FGF2 can inhibit activation of CFs and analysed the regulatory effects of FGF2 on CFs at the gene level. We identified a set of DEGs that are systematically regulated by FGF2, including those related to matrix protein, effector, membrane receptor, actin filament, signalling protein and nucleoprotein (Fig. 8A). GSEA results reveal that FGF2 can suppress ECM organization, actin filament and cell-substrate adhesion, suggesting FGF2 inhibits the differentiation of CFs into myofibroblasts (Fig. 8B). In addition, FGF2 increased axon guidance, transport vesicle and basolateral plasma membrane functions (Fig. 8B).

Figure 8 Summary of regulated genes and functions in CFs by FGF2.

(A) Subcellular location of proteins of DEGs indicated as gene symbols. (B) Summary of regulated functions in CFs by FGF2. The purple and red shapes indicate ECM proteins, and the navy blue shapes indicate the effectors regulating fibroblast activation. a-SMA, alpha smooth muscle actin; CTHRC1, collagen triple helix repeat containing 1; CTGF, connective tissue growth factor; TGF-, transforming growth factor-.

Previous studies have shown that recombinant FGF2 (LMW-FGF2) can inhibit activation of fibroblasts from a variety of tissues and organs, including the heart, lung, skin, etc., suggesting FGF2 is a promising protein drug for regulation of fibroblasts in tissue repair (Dolivo, Larson & Dominko, 2017; Koo et al., 2018; Liguori et al., 2018; Maltseva et al., 2001; Santiago et al., 2014b). Svystonyuk et al. (2015) and Liguori et al. (2018) found that FGF2 could suppress the expression of α-SMA and collagen genes, preventing the differentiation of CFs into myofibroblasts and exhibiting anti-fibrotic effects. Though the suppressive effect of FGF2 on CFs activation was confirmed in the previous studies, only a few specific genes or proteins were studied, e.g., α-SMA, collagens and matrix metallopeptidases. In this study, transcriptome sequencing was used to investigate the regulation of gene expression. Our findings indicate that FGF2 could regulate transcription of a large number of ECM-relevant genes, including collagens, elastin (Eln), periostin (Postn) and glypicans, which are involved in the progression of cardiac fibrosis. Matrix metallopeptidase 16 (Mmp16) and chondroitin sulfate proteoglycan 4 (Cspg4) were inversely increased, and it was found MMP16 could interact with CSPG4 to degrade collagen type I (Iida et al., 2001).

RNA sequencing technique had been utilized to investigate the effects of FGF2 on mRNA transcription in fibroblasts derived from the skin and lung (Fortier et al., 2021; Kashpur et al., 2013; Wu et al., 2021a). As described, the expression of cytoskeleton genes and ECM organization genes, e.g., Acta2, Myh10, Eln, Col1a1, Col1a2 and Postn, was significantly decreased, which is consistent with our findings (Fortier et al., 2021; Kashpur et al., 2013; Wu et al., 2021a). Nevertheless, fibroblasts isolated from different tissues exhibit certain heterogeneity, which may result in the diversity of gene expression regulated by FGF2.

CFs could be activated by multiple factors, such as extracellular effector proteins and mechanical stress, to overexpress profibrotic genes, resulting in fibrosis. Many genes of secreted effector proteins were decreased by FGF2, in which Tgfb2, Tgfb3 and Ctgf were found to play critical roles in stimulating tissue fibrosis (Lipson et al., 2012; Sun et al., 2021; Vainio et al., 2019). The decreased membrane receptors, Agtr1a and Fgfr3, were also reported to promote fibrosis (Chakraborty et al., 2020; Czepiel et al., 2022; Tian et al., 2020). Shimizu et al. (2017) found that deletion of Rock2 reduced expression of connective tissue growth factor (CTGF), α-SMA and FGF2, which attenuated cardiac hypertrophy and fibrosis. Our sequencing results indicate that Rock1 and Rock2 were both decreased by FGF2. Thus, FGF2 might be able to suppress cardiac fibrosis by multiple signalling pathways.

Although mRNA sequencing has shown the regulatory function of FGF2 on CFs gene expression, the underlying mechanism of signalling transduction is not fully elucidated. It was found that FGF2 can active both FGF receptor 1 (FGFR1) and FGFR3, but further studies are required to understand the effects of FGFRs and their downstream signalling on fibroblast activation and gene expression (Sun et al., 2022; Zhang et al., 2006). Proteomics could provide additional insights into signal transduction pathways, which could illuminate the regulatory mechanism of FGF2 on CFs gene expression.

In conclusion, our study showed that FGF2 could suppress CFs activation and ECM expression, suggesting FGF2 is a potential antifibrotic growth factor in the treatment of cardiac injury. The identification of DEGs in CFs, regulated by FGF2, provides additional references for basic research and clinical transformation of FGF2 in treating tissue fibrosis.

Supplemental Information

Supplemental Information 1 Correlation determination of RNA sequencing data.

The values of Pearson correlation Coefficient are displayed. CFs-CT: CFs-CON; CFs-FGF2: CFs-FGF2.

Click here for additional data file.

Supplemental Information 2 GO enrichment analysis of membrane organization relevant genes.

(A): Display of genes related to membrane microdomain, basolateral plasma membrane, Golgi membrane, transport vesicle and synaptic membrane. (B): GSEA plots of basolateral plasma membrane and transport vesicle functions.

Click here for additional data file.

Supplemental Information 3 GO enrichment analysis of actin cytoskeleton and cell shape relevant genes.

(A): Display of genes related to actin cytoskeleton, apical part of cell, cell leading edge and site of polarized growth. (B): GSEA plots of actin cytoskeleton and axon guidance.

Click here for additional data file.

Supplemental Information 4 Primers for RT-PCR in this study.

Click here for additional data file.

Supplemental Information 5 Differentially expressed genes.

Click here for additional data file.

Additional Information and Declarations

Competing Interests

Author Contributions

Animal Ethics

Data Availability

The authors declare that the research was conducted in the absence of any commercial or financial relationships that could be construed as a potential conflict of interest.

Changye Sun conceived and designed the experiments, performed the experiments, analyzed the data, prepared figures and/or tables, authored or reviewed drafts of the article, and approved the final draft.

Mengru Bai performed the experiments, analyzed the data, prepared figures and/or tables, authored or reviewed drafts of the article, and approved the final draft.

Yangyang Jia performed the experiments, authored or reviewed drafts of the article, and approved the final draft.

Xiangqin Tian performed the experiments, authored or reviewed drafts of the article, and approved the final draft.

Yonglong Guo performed the experiments, authored or reviewed drafts of the article, and approved the final draft.

Xinhui Xu analyzed the data, authored or reviewed drafts of the article, and approved the final draft.

Zhikun Guo conceived and designed the experiments, authored or reviewed drafts of the article, and approved the final draft.

The following information was supplied relating to ethical approvals (i.e., approving body and any reference numbers):

Xinxiang Medical University Animal Supervision Committee (approval no. XYLL-2020078)

The following information was supplied regarding data availability:

The raw sequence files for both CFs-CON and CFs-FGF2 are available at NCBI Sequence Read Archive: PRJNA833249.

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
