# Peer review of "mRNA sequencing reveals the distinct gene expression and biological functions in cardiac fibroblasts regulated by recombinant fibroblast growth factor 2"

_PeerJ, doi:10.7717/peerj.15736_

## Round 0.1 · original submission · Major Revisions

Before I send this paper out for peer review, could you please address the following issues:

Each figure should state clearly how many biological and technical replicates have been performed for each component of the figure.

Please provide more information on RNAseq. I refer you to our guidelines:

> Studies that use RNA-seq data analysis to examine transcription patterns must consider sequencing depth and the number of biological replicates. As noted in 'A survey of best practices for RNA-seq data analysis' the number of replicates that should be included in an RNA-seq experiment depends on both the technical variability in the RNA-seq procedures and the biological variability of the system in question. These submissions must include both (i) a power analysis calculation and (ii) information on biological and technical replicates used to achieve the claimed statistical power. Submissions that do not report sufficient replication will not be considered.

Once this has been satisfactorily addressed I will then send this out for peer review.

+++++
- Please edit your manuscript to include **_BOTH_**:
1. a **_power analysis calculation_** (software tools are available, e.g. <https://doi.org/doi:10.18129/B9.bioc.RNASeqPower> (see [usage instructions](https://bioconductor.org/packages/release/bioc/vignettes/RNASeqPower/inst/doc/samplesize.pdf>))) **_AND_**
2. **_information on biological and technical replicates_** used to achieve the claimed statistical power

- A suitable wording for the power analysis would be something like: '_The statistical power of this experimental design, calculated in_ RNASeqPower _is_ 0.84' (replace the program name if you used a different tool and provide the actual calculated power).
- An online implementation of RNASeqPower is available at <https://rodrigo-arcoverde.shinyapps.io/rnaseq_power_calc/> if you prefer not to use the R command line.
+++++

---

## Round 0.2 · Major Revisions

As requested previously:

Each figure should state clearly how many biological and technical replicates have been performed for each component of every figure. Please amend and resubmit.

---

## Round 0.3 · Minor Revisions

As you will see from their comments, all three reviewers are positive in tone about your work, but each highlights an extensive list of suggestions/corrections/clarifications. Please address all of these points in your revised paper, and outline in a cover letter your response and the changes you have made. I look forward to seeing the revised paper in due course.

Reviewer 1 ·

Basic reporting

See additional comments

Experimental design

See additional comments

Validity of the findings

See additional comments

Additional comments

In this manuscript, Sun et al. describes transcriptomic effects of FGF2 stimulation in cardiac fibroblasts.
The work is fairly straightforward and mostly descriptive in nature, but otherwise interesting and of potentially broad applicability, due to conserved effects of FGF2 recognized in mesenchymal cells of different origins.
Since the work is mostly descriptive, the authors must do a better job of placing their findings in the greater context of the literature. More information and other specific points below:
The manuscript contains many small errors and could use proofreading and editing by a fluent speaker. For example, in the abstract alone, “secret” should be “secrete”, “was found FGF2 could” should be “was found that FGF2 could”, etc.
Genes should be given their appropriate names and be italicized.
In the introduction, the authors mention that FGF2 has been described to inhibit activation of fibroblasts from several organs, but authors should also note other studies that have performed transcriptomic comparisons of fibroblasts stimulated with FGF2 such as Kashpur et al. 2013 BMC Genomics “FGF2-induced effects on transcriptome associated with regeneration competence in adult human fibroblasts” Wu et al. 2021 PeerJ “RNA sequencing analysis of FGF2-responsive transcriptome in skin fibroblasts” Xuan et al. 2016 J. Dermatol. Sci. “The activation of the NF-kB-JNK pathway is independent of the PI3K-Rac1-JNK pathway involved in the bFGF-regulated human fibroblast cell migration”, Wang et al. 2017 Front. Pharmacol. “Feedback activation of basic fibroblast growth factor signaling via the Wnt/beta-catenin pathway in skin fibroblasts.” and Fortier et al. 2021 JCI Insight “Myofibroblast dedifferentiation proceeds via distinct transcriptomic and phenotypic transitions.,” as well as in other cell types including endothelial cells and VICs (see summary table in Dolivo 2022 J. Mol Med. “Anti-fibrotic effects of pharmacologic FGF-2: a review of recent literature”). Authors should compare and contrast their results to the results of these other groups’ analyses. Do authors find that key DEGs are conserved in the FGF2 responses of fibroblasts from these different experiments? These are likely signs of conserved fibroblast responses. Do authors also find differences in the DEGs of cardiac fibroblasts? These could be tissue-specific differences. These would be interesting to note and talk about in the discussion, as they relate to future directions for research. It is not currently well understood how FGF2 might differentially effects mesenchymal cells from different organs, and what the upstream factors regulating these differential responses might be (differential expression of FGFR gene expression, or of isoform expression of these genes? Epigenetic factors? Etc.).
Authors note that FGF2 has been applied (preclinically and clinically) to promote regeneration in skin wounds, but there is also experimental evidence in animal models of vocal fold scarring, pulmonary fibrosis, hepatic fibrosis, etc. There is a fairly substantial literature of anti-fibrotic FGF-2 of cellular and animal models already in existence, and authors should at least acknowledge this. If anything, it probably lends greater importance to their data, due to the translatability of fibroblast responses across tissues.
Centrifugation speeds should be in x g.
Were cells for immunofluorescence cultured on glass coverslips? Or just in tissue culture plates?
Concentrations of antibodies used for immunofluorescence should be listed.
What construction of the rat reference genome was used for mapping of samples?
Were p values for RNA-seq experiments adjusted for multiple comparisons?
What are the models of the microscopes used for these experiments?
Line 161 “filed” should be “field”
Authors should justify/explain their choice of Ccdc80 as a myofibroblast-associated ECM gene. I am quite familiar with the myofibroblast/fibrosis literature, and I have never heard of this gene before.
The gene that encodes CTGF is Ccn2, not Ctgf.
For readers who aren’t particularly familiar with transcriptomics, it might be nice to include the 2D PCA output by Deseq2 just to show groupings of the RNA-seq samples as a supplementary figure or as a panel in figure 5.
What was the basis for the choice of genes in Fig. 5E?

Line 286 Tgfb2 and Tgfb3
Authors should expand the discussion section with much discussion about further unanswered questions, and better place their own findings within the greater context of the anti-fibrotic FGF2 literature.
Authors should also describe the limitations of their own work. They only looked at transcript expression (with the exception of a couple of proteins), they did not characterize receptor or isoform expression of FGFR in their cells, and so they do not know which receptors drive the FGF2 response in these cells, they do not know which downstream pathways regulate these different classes of DEGs (FGF2/FGFR can activate ERK, JNK, p38, PLCgamma, and other pathways somewhat independently), as well as other limitations.

·

Basic reporting

This report provides a review of an article, which is written in professional English and is generally clear in conveying its message. However, the abstract (lines 19-23), could benefit from a more comprehensive description of the results, including a summary of the GO analysis. Additionally, in the introduction (lines 62-65), the logic relating to HMW-FGF2 promoting myocardial hypertrophy, while only LMW-FGF2 has a protective effect, needs clarification. It must be made clear why FGF2 serves as an effective target in cardiac myocytes. Specifically, it should be stated that recombinant/LMW-FGF2 may be the effective target, rather than FGF2 itself.

The article provides a comprehensive literature review and a thorough background introduction. However, the logical relationship between CF, SMA, and other factors (lines 27-31) could be improved through rewriting for greater clarity.

The structure of this article is sound, and both the figures and raw data sharing are professional. Nevertheless, it should be noted that in Figure 8b, the purple and red proteins are extracellular proteins, not intracellular ones as they appear to be entering the cell membrane in the diagram.

Overall, this article is self-contained and presents results relevant to the hypothesis. Nonetheless, future research should further refine the article's language, and rigorously improve experimental design, data processing, and figure presentation.

Experimental design

This article evaluates the experimental design of an original primary research article. The aim and scope of this article are consistent with those of the journal. The research question is clear, and the authors have used a series of experiments, including in vivo murine experiments, in vitro cardiac myocyte cultures, IF staining, mRNA sequencing, to address the research question step by step. The authors' use of these methods is technically sound and ethically appropriate.

The method section of the article is clearly described, which facilitates replication by other researchers. Overall, the experimental design is well-conceived and executed with a high degree of rigor, ensuring that the results are reliable and valid. The approach taken provides a comprehensive understanding of the research question and contributes significantly to the body of knowledge in the field.

In conclusion, the experimental design of this article is strong, reflecting good scientific practice and contributing significantly to the field. The authors have successfully addressed the research question using appropriate methods in compliance with ethical standards. This study provides a valuable contribution to the field and sets a good example for future research.

Validity of the findings

This article evaluates the validity of the findings in an article. The core findings of this article are sufficiently impactful and novel. At the mRNA level, the article provides a detailed explanation of FGF2's role in preventing myocardial fibrosis. However, there are some areas where further improvement can be made to ensure the validity of the findings.

Firstly, while the results of mRNA sequencing are promising, it is recommended that the authors conduct RT-PCR validation of several representative genes to confirm their findings.

Secondly, some experimental results lack sufficient data support, such as "Regulation of CFs activation by serum" in Figure 2. This part of the article does not provide convincing evidence for its conclusion. It is advisable that the authors either provide additional data to support this finding or omit this part from the results section.

Thirdly, throughout the article, the conclusion and discussion do not differentiate clearly between recombinant FGF2 (i.e., LMW-FGF2) and natural FGF2. This also includes HMW-FGF2, which may lead to confusion. It is important that the authors be more cautious and precise when discussing the conclusions related to FGF2.

In conclusion, while the core findings of this article are significant, there are some areas where further improvement could enhance the validity of the findings. It is recommended that the authors conduct RT-PCR validation of their mRNA sequencing results, provide stronger data support for their findings, and use clear language in discussing the conclusions related to FGF2.

Additional comments

This article contains valuable information, but there are some minor issues that require attention.

Firstly, the author uses several abbreviations without providing explanations at first use. For example, SMA on line 17 and paraformaldehyde (PFA) on line 169. Additionally, Figure 8a contains many proteins, and it would be helpful if the author could provide the abbreviations. A brief explanation of this figure would be helpful.

Secondly, on line 264, the last word "a" is in a different font than the rest of the article. It is recommended that the author correct this formatting error.

Overall, these issues do not significantly detract from the quality of the article or its findings. However, correcting these minor problems would help readers better understand the content and appreciate the research presented in the article.

·

Basic reporting

Some superfluous information e.g Figure 4.
Until reading the eMethods the experimental design was not clear.

Experimental design

Typical experimental design, just some concerns about how the RNAseq data was filtered.

Validity of the findings

The authors present appropriate data, but I am not clear of some of the potential consequences as groups of genes are regulated in both directions.

Additional comments

Using isolated neonatal cardiac fibroblasts, the authors investigated the action of FGF2 by RNAseq analysis.
• What was the basis selecting the different concentrations of FGF2 and the selection for the 30ng/ml
• In the introduction two subtypes of FGF2 were introduced high molecular weight FGF2 (HMW-FGF2) and low molecular weight FGF2 (LMW-FGF2) that had contrasting effects in the heart. In this study, I am not sure which subtype was used experimentally.
• Can the authors confirm their filtering strategy my principle concern is the p<0.05 that doesn’t seem to take account of multiple testing, this should be either an adjusted p value of a false discovery rate.
• In addition, it seems fold change has also been used for filtering,
o fold change >0.75 or fold change < -0.75, surely this is log 0.75 ie a fold change of approximately +/- 1.68)?
• I am not sure of the value of log10(basemean) > 2, is this to remove genes that are expressed at low levels across the experiment? If so why log10(basemean) > 2?
• Figure 1 legend, P0-P3, not P5
• I would suggest low serum is 0.1%, I would also suggest that comparing 2.5%, 5% and 10% serum is highly subjective from the data presented. Although I agree 2.5% seems like a reasonable serum concentration for these experiments, but not necessarily for the reasons the author suggest.
• Is there a reason why the 100ng/ml group is not shown in figure 3A.
• Can appropriate measures of significance, taking account of multiple testing, be added to Figure 3B. This may help the decision why 30ng/ml was used.
• Was alpha SMA measure by RT-qPCR?
• One of the issues of the delta delt Ct method and presenting the data like this on the same graph is that all of the genes tested look as though they are all expressed at the same level, which is unlikely, can the authors provide Ct values to the reviewers? This applies to Figure 3B and 5E, significance must be added to the graphs.
• Would the authors agree the myofibroblast markers are more sensitive to FGF2 than the extracellular membrane markers, is this biologically relevant?
• Can the authors explain the significance of Figure 4 highlighting highly expressed genes, I accept some are regulated, but otherwise not relevant as there is no indication of significance and the biological relevance is not clear, in fact confuses the issue.
• How do the activation and inhibition of genes in the various groups you highlight contribute to phenotype ie is it clear there is activation of ECM when many of the genes are regulated in opposite directions.
• How do the authors suggest targeting FGF2 to cardiac fibroblasts in the whole organisms in the disease setting?

---

## Round 0.4 · accepted · Accept

Thanks for addressing the comments from the previous review. I and the reviewers are satisfied with these changes and I therefore am recommending acceptance.

Reviewer 1 ·

Basic reporting

N/A

Experimental design

N/A

Validity of the findings

N/A

Additional comments

Authors have adequately addressed my initial queries and I have no further concerns

·

Basic reporting

See additional comments

Experimental design

See additional comments

Validity of the findings

See additional comments

Additional comments

I have thoroughly assessed the manuscript titled "mRNA sequencing reveals the distinct gene expression and biological functions in cardiac fibroblasts regulated by recombinant fibroblast growth factor 2." The authors have presented a well-executed study that addresses the gap in knowledge concerning the regulatory role of FGF2 on cardiac fibroblasts (CFs) and its potential therapeutic role in treating cardiac fibrosis.

The methodology employed in this study is robust and comprehensive. The use of RNA sequencing to analyze mRNA expression in CFs and identify differentially expressed genes regulated by FGF2 is highly appropriate. The inclusion of gene ontology analysis and gene set enrichment analysis (GSEA) further strengthens the study's findings and provides a deeper understanding of the biological functions of the differentially expressed genes.

It is noteworthy that the authors have identified 430 up-regulated genes and 391 down-regulated genes regulated by FGF2. Their findings provide valuable insights into the potential therapeutic role of FGF2 in treating cardiac fibrosis. The analysis of the differentially expressed genes and their association with ECM organization, cell adhesion, actin filament organization, and axon guidance is well-presented and thoroughly discussed.

The authors have also demonstrated the consistency between the regulatory functions of FGF2 and the findings of the gene enrichment analysis through further cellular experiments. This effort further validates the results of the study and adds credibility to the conclusions drawn.

In conclusion, I believe that this study makes a significant contribution to the field, and the authors have presented their findings clearly and persuasively. The study establishes a strong foundation for future research aimed at uncovering the underlying mechanisms of CFs gene expression regulated by FGF2. I recommend that this manuscript be accepted for publication without further revisions.